# Dummy Template-Based Molecularly Imprinted Membrane Coating for Rapid Analysis of Malachite Green and Its Metabolic Intermediates in Shrimp and Fish

**DOI:** 10.3390/molecules28010310

**Published:** 2022-12-30

**Authors:** Yi Zhang, Shaofeng Li, Yurong Gu, Jianying Zhang, Zhenfeng Yue, Liao Ouyang, Fengjuan Zhao

**Affiliations:** 1School of Materials and Environmental Engineering, Shenzhen Polytechnic, Shenzhen 518055, China; 2Food Inspection & Quarantine Center, Shenzhen Customs, Shenzhen 518045, China

**Keywords:** malachite green molecularly imprinted membranes, template analogs malachite green, colloidal gold strip, residuals analysis

## Abstract

A novel malachite green molecularly imprinted membrane (MG-MIM) with specific selectivity for malachite green (MG) and leucomalachite green (LMG) was prepared using a hydrophobic glass fiber membrane as the polymer substrate, methyl violet as a template analog, 4-vinyl benzoic acid as the functional monomer, and ethyleneglycol dimethacrylate as the crosslinking agent. MG-MIM and non-imprinted membrane (NIM) were structurally characterized using scanning electron microscopy, surface area analyzer, Fourier-transform infrared spectrometer and synchronous thermal analyzer. The results showed that MG-MIM possessed a fluffier surface, porous and looser structure, and had good thermal stability. Adsorption properties of MG-MIM were investigated under optimal conditions, and adsorption equilibrium was reached in 20 min. The saturated adsorption capacities for MG and LMG were 24.25 ng·cm^−2^ and 13.40 ng·cm^−2^, and the maximum imprinting factors were 2.41 and 3.20, respectively. Issues such as “template leakage” and “embedding” were resolved. The specific recognition ability for the targets was good and the adsorption capacity was stable even after five cycles. The proposed method was successfully applied for the detection of MG and LMG in real samples, and it showed good linear correlation in the range of 0 to 10.0 μg·L^−1^ (R^2^ = 0.9991 and 0.9982), and high detection sensitivity (detection limits of MG and LMG of 0.005 μg/kg and 0.02 μg·kg^−1^ in shrimp, and 0.005 μg/kg and 0.02 μg/kg in fish sample). The recoveries and relative standard deviations were in the range of 76.31–93.26% and 0.73–3.72%, respectively. The proposed method provides a simple, efficient and promising alternative for monitoring MG and LMG in aquatic products.

## 1. Introduction

Malachite green (MG) has been widely utilized as an industrial dye, in the chemical preparation of bactericidal materials, and in aquaculture as a fungicide for fish [1,2,3]. MG is a toxic triphenylmethane chemical. Moreover, its metabolite in aquatic organisms is leucomalachite green (LMG), which is more toxic than MG and has been recognized as a Class 2 carcinogen by the International Agency for Research on Cancer [4,5]. Unfortunately, due to its abundance, low cost and high efficiency, MG has been occasionally abused in fish production for human consumption. To date, most countries have established strict restrictions to control chemical residues in aquatic products. Additionally, countries such as the United States, China, those in Europe, and some Southeast Asian countries have placed a ban on MG to treat marine ailments. According to EU legislation, the total amount of MG and LMG residues permitted in animal-derived foods is limited to 2 μg·kg^−1^. Japan also clearly stipulated that MG cannot be detected in imported aquatic products. Therefore, the detection of MG and LMG in the environment is very important [6,7,8,9].

In recent years, a variety of analytical methods have been used to determine MG content, including liquid chromatography (LC) coupled with visible/fluorescence detection, LC coupled with mass spectrometry (LC-MS), and electrochemical detection [9,10,11,12]. In general, chromatography and its combined techniques are sensitive methods for measuring MG and LMG. Gas chromatography–tandem mass spectrometry (GC-MS/MS) is often used to detect LMG but not MG [10]. Although chromatography-fluorescence spectrophotometry can detect fluorescent substances with high sensitivity, MG has no fluorescence properties. Therefore, during the pre-treatment process, MG must be reduced to LMG in order to be detected. UV–visible spectrophotometry (UV–Vis) can determine substances with characteristic absorption peaks in the UV–visible range. However, there is a high level of interference adjacent to the characteristic peaks of MG and LMG, making it impossible to accurately quantify the target objects [11].

In the case of complex sample matrix and low concentration of residues, specific sample preparation techniques are utilized, such as organic solvent extraction or solid phase extraction (SPE), and concentration and purification using SPE column. However, traditional SPE materials and immunoaffinity chromatography materials often exhibit various shortcomings. Currently, available SPE columns, including HLB columns and C_18_ monolithic columns, are limited by their adsorption selectivity and enrichment efficiency. Furthermore, this approach relies on a high ratio of solvent, causing poor compatibility with colloidal gold immunization strip, which allows for the inactivation of specific reactions related to the antigen and antibody. Teepoo et al. developed a rapid and simple immunochromatographic strip test assay based on competitive format for LMG detection. The method was completed in 5 min without any incubation, washing or blocking steps required, and had a detection limit of 2 µg·L^−1^ [12]. However, in real sample detection, the ‘matrix effect’ in the sample reduces the performance of the test strip. Therefore, it remains a considerable challenge to develop a simple and rapid method for MG and LMG pretreatment in real samples.

In 1972, Wuff et al. prepared a type of molecularly imprinted polymers (MIPs) with chiral separation, which was successfully used in the separation of amino acids. This research inspired the development of molecular imprinting technology (MIT) [13]. In recent years, MIPs have attracted extensive attention due to their excellent selectivity, high adsorption efficiency, good thermal stability and simple preparation methods. They have been widely applied in many fields, such as SPE and sensor research [14,15,16]. Molecularly imprinted membrane (MIM) combines the specific recognition of MIPs for a specific imprinted molecule, while harnessing the simple operation and mild conditions of membrane separation [17,18,19,20,21]. Therefore, MIM exhibits desirable characteristics, such as high selectivity and continuous operation. Its specific recognition mechanism involves the template molecule being preferentially adsorbed by MIM because it matches the imprinting site on MIM, while hindering the specific adsorption between the interferent and MIM. Additionally, the binding ability of MIM surface imprinting site for the target largely determines the specific adsorption ability of MIM for the target [22,23,24].

Nowadays, a variety of imprinting strategies have been reported for the preparation of MIPs [25,26]. The dummy template strategy for the preparation of MIPs selects template substances with the same or a similar structure as the target. This approach solves the problems of expensive or inaccessible templates and avoids false positives [27]. Xue used 2,6-dichloroindole-4-chloroimine (DBQ) as the dummy template for the analysis of trace 2,6-dichloroindole-1,4-benzoquinone (2,6-DCBQ) in water samples. This method effectively reduced the toxicity and cost during synthesis, avoided the quantitative inaccuracy caused by template leakage, and achieved high adsorption selectivity for targets [28]. Sadat designed a highly selective MIP to attenuate biofilm formation of multidrug-resistant pathogen pseudomonas aeruginosa by trapping pseudomonas quinolone signaling (PQS). However, considering the low thermal stability and high price of standard PQS, alternative molecules with similar imprinting sites were synthesized as pseudo-templates to prepare the desired MIPs [29].

In this work, a glass fiber membrane was employed as the substrate and methyl violet (MV) as the dummy template molecule, allowing the synthesis of MG-MIM by surface molecular imprinting. In order to obtain MG-MIM with excellent imprinting effect and high adsorption capacity, the influences of process conditions were systematically studied. A schematic diagram of MG-MIM preparation is shown in Figure 1. The morphology, internal structure, molecular composition and thermal stability of MG-MIM were investigated using scanning electron microscopy (SEM), Brunauer–Emmett–Teller (BET) surface area, Fourier-transform infrared spectrometry (FT-IR) and thermal gravimetric analysis-differential scanning calorimetry (TGA-DSC). The as-prepared MG-MIM selectively adsorbed MG and LMG. Through the combination of MIT with rapid detection technology, MIM exhibited accurate target recognition, and effectively improved inaccurate detection results caused by a lack of appropriate pre-processing in the existing rapid detection technology.

## 2. Results and Discussion

### 2.1. Synthesis of MG-MIM

In order to prepare a material that could specifically adsorb MG and its metabolite LMG, hydrophobic glass fiber membrane was selected as the substrate material. The optimal synthesis conditions were determined through a single factor investigation. MG-MIM was prepared using hydrophobic glass fiber membrane as the polymer substrate, MV as the template substitute (Appendix A), 4-vinyl benzoic acid as the functional monomer (Appendix A), ethyleneglycol dimethacrylate (EGDMA) as the crosslinking agent, azobisisobutyronitrile (AIBN) as the initiator, and acetonitrile (ACN) as the solvent (Appendix A). The molar ratio of MV, 4-vinylbenzoic acid and EGDMA was 1:4:24 (Appendix A), and the polymerization time was 16 h (Appendix A). MG-MIM displayed excellent specific selectivity for MG and LMG. The adsorption concentrations for MG and LMG were 6.27 ng·mL^−1^ and 2.46 ng·mL^−1^, and the imprinting factors were 3.13 and 2.70, respectively.

### 2.2. Structural Characterization of the MIP

#### 2.2.1. SEM Analysis

In this study, scanning electron microscope (SEM) was used to examine the surface structure and morphological characteristics of the as-prepared MG-MIM and NIM. Figure 2A,B shows the surface morphologies of MG-MIM and NIM at 10 K magnification, respectively. The results showed that NIM had a smooth and dense surface, whereas the surface of MG-MIM was porous and loose. Figure 2C,D shows the surface morphologies of MG-MIM and NIM at 20 K magnification, respectively. The images revealed that the surface of MG-MIM was fluffy, the pore structure was evenly distributed, and the pore size was uniform. In general, the as-prepared MG-MIM material formed a porous structure on the surface and provided adsorption sites for the target [27]. Compared with NIM, MG-MIM had a higher transfer rate, and the defect of embedding at the imprinted sites was improved.

#### 2.2.2. Specific Surface Area Analysis

In addition, the BET specific surface area and pore structure of as-prepared MG-MIM were determined [28], and the results are shown in Figure 3. When the relative pressure was between 0.00 and 0.06, the adsorption capacity of MG-MIM showed a rapid increase. When the relative pressure was between 0.06 and 0.70, the adsorption capacity of MG-MIM surface only showed a gradual increase. At relative pressure between 0.70 and 1.00, the adsorption capacity of MG-MIM surface increased rapidly. Furthermore, a hysteresis loop formed between the desorption curve and the adsorption curve, which indicated the presence of interconnected channels in MG-MIM. Hence, the as-prepared MG-MIM contained a certain amount of porous structure. According to the obtained BET results, the specific surface area of MG-MIM was 473.5 m^2^·g^−1^ and the pore volume was 0.37 cm^3^·g^−1^, indicating its strong adsorption capacity.

#### 2.2.3. FT-IR Analysis

As shown in Figure 4, MG-MIM displayed an absorption peak at 3342.64 cm^−1^, which corresponded to the stretching vibration of intermolecular hydrogen bonds, and indicated the presence of hydrogen bonding in MIP [23]. The peak at 3001.24 cm^−1^ corresponded to the stretching vibration absorption of C-H from =C-H and benzene. The peaks at 2912.51 cm^−1^ and 2613.55 cm^−1^ corresponded to the stretching vibration absorption peaks of O-H bond from the carboxyl group. The peak at 1701.21 cm^−1^ corresponded to the stretching vibration absorption peak of C = O bond from the carboxyl group. The peak at 1512.19 cm^−1^ was attributed to the vibration absorption peak of benzene [29]. The peak at 1402.25 cm^−1^ corresponded to the bending vibration absorption peak of C-H bond from EGDMA methyl. The peak at 1269.16 cm^−1^ corresponded to the stretching vibration absorption peak of C-O-C ester bond. The peak at 1014.55 cm^−1^ corresponded to the vibration absorption peak of C-H bond bending over the benzene ring. The peak at 950.90 cm^−1^ represented the vibration absorption peak of O-H bond bending over the carboxyl group. The results showed that the majority of the functional monomer 4-vinyl benzoic acid was crosslinked with EGDMA. FT-IR spectra of MG-MIM and NIM revealed a similar chemical skeleton structure, indicating that the substitution of the template molecules did not change the chemical composition of the polymer, but changed the arrangement of 4-vinylbenzoic acid in the polymer structure via hydrogen bonding and π–π conjugation [30].

#### 2.2.4. TGA-DSC Analysis

The thermal stability of MG-MIM was investigated using TGA-DSC [31,32,33,34]. Figure 5 shows that as the temperature increased, MG-MIM exhibited almost no thermal weight loss up to 120 °C, highlighting its stability. At 280 °C, the thermal weight loss of MG-MIM was 18.02%, which may stem from the volatilization of excess monomers and crosslinking agents in the polymerized system, as well as the decomposition of the polymer that occurred over time. The thermogravimetric loss of MG-MIM between 280 and 460 °C was 81.31%, which was caused by combustion decomposition of polymerization layer. At 460 °C, the polymer was completely decomposed and the polymer film volatilized rapidly.

### 2.3. Adsorption Properties of the MIP

#### 2.3.1. Adsorption Isotherms

The static adsorption of MG-MIM and NIM was studied, and the concentration range of the target substance was 0–30 ng·mL^−1^. The results shown in Figure 6 revealed that within the concentration range of 0–22 ng·mL^−1^, the adsorption capacity of MG-MIM and NIM for the target substance increased with the increase in concentration of the target substance. When the concentration exceeded 22 ng·mL^−1^, the adsorption capacity of MG-MIM and NIM for the target substance plateaued, reaching a static adsorption equilibrium. At this time, the equilibrium adsorption capacities of MG-MIM for MG and LMG were approx. 24.25 ng·cm^−2^ and 13.40 ng·cm^−2^. In contrast, the equilibrium adsorption capacities of NIM for MG and LMG were approx. 10.33 ng·cm^−2^ and 4.24 ng·cm^−2^. The maximum imprinting factors for MG and LMG were 2.41 and 3.20, respectively. Therefore, the obtained results indicated that MG-MIM had good specific adsorption and recognition performance towards MG and LMG.

#### 2.3.2. Adsorption Kinetics

The dynamic adsorption of MG-MIM and NIM was studied, where the adsorption kinetics curve (Figure 7A,B) showed that the adsorption rate of MG-MIM for MG and LMG increased significantly faster than that of NIM within 0–20 min. At 20 min, the adsorption of both materials reached saturation. The adsorption capacities of MG-MIM for MG and LMG were approx. 23.24 ng·cm^−2^ and 15.22 ng·cm^−2^, and those of NIM for MG and LMG were approx. 10.13 ng·cm^−2^ and 3.90 ng·cm^−2^, respectively. During the adsorption process, the adsorption capacity of MG-MIM for the target was significantly higher than that of NIM due to the addition of an alternative template molecule MV, which had a similar structure as the target in the preparation process of MG-MIM. After elution of the template, three-dimensional holes matching the structure and size of MV remained in the imprinted membrane [35]. The pore contained interaction sites that complemented the functional groups of MV, and these action sites recognized their structural analogs, MG and LMG, allowing specific binding. Additionally, MG-MIM had a relatively fast response ability for both MG and LMG, and could reach adsorption equilibrium within 20 min.

#### 2.3.3. Interference, Stability and Reproducibility Studies

In order to illustrate the specific adsorption ability of MIM for the target molecules, MG-MIM was used to extract a single component from a mixed solution of basic green, basic red 9, MG and LMG, respectively. As shown in Figure 8, at the same standard level of 10 ng·mL^−6^, MG-MIM specificity towards MG and LMG adsorption was greater than that for basic green and basic red 9, indicating that MG-MIM could specifically adsorb MG and LMG.

In order to further explain that the recognition effect on MIM was mainly caused by specific adsorption of the target by the imprinting site, the extraction experiment of the mixed solution of basic green, basic red 9, MG and LMG was carried out using MG-MIM. As shown in Figure 9, the adsorption capacity of MG-MIM for MG and LMG was much higher than that of basic green and basic red 9, which indicated that the adsorption ability of MG-MIM for MG and LMG was specific adsorption. Furthermore, the adsorption capacity of MIM for MG and LMG did not show a significant decrease. Hence, addition of basic green and basic red 9 had no obvious effect on the specific adsorption of the imprinted membrane. The results also revealed that MIM prepared in this study had stable adsorption capacity for the target substance.

The effects of acetonitrile and pH on the adsorption capacity of MG-MIM were further investigated, as shown in Appendix A. The results are listed in Appendix A. The as-prepared MG-MIM exhibited excellent acid resistance, alkali resistance and organic solvent resistance (RSD < 10%). The RSD for ten repeated measurements of MG-MIM was smaller than that of NIM.

MG-MIM samples were reused for ten times to evaluate their extraction stability. The adsorption concentration of MG-MIM for MG and LMG is shown in Appendix A. The adsorption capacity of MG-MIM showed a slight and gradual decrease with the increase in its repeated use. After the 5th use, the adsorption capacity of MG- MIM for MG was 91% of the first time, while that for LMG was 87% of the first time. The adsorption capacity of MG-MIM for the target was not significantly attenuated, and it still maintained good specificity. After the 6th use, the adsorption capacity of MG-MIM for MG was 82% of the first time, but that for LMG was only 71% of the first time. From the 6th to the 10th extraction, the adsorption capacity of the same imprinted membrane for LMG and MG began to decline significantly. The reason may be that after multiple adsorption elution processes, some imprinting sites may be damaged during the elution process. Therefore, the experimental results indicate that the adsorption capacity of MG-MIM was stable during five cycles.

### 2.4. Evaluation of Analytical Methods

#### 2.4.1. Evaluation of Instrument Method

##### Linearity Range, Limit of Detection (LOD), and Limit of Quantification (LOQ) Determination

A calibration curve of MG and LMG was constructed by plotting peak area versus MG and LMG concentration in the range of 0–50.0 µg·L^−1^. According to the principle of stepwise dilution and traditional SNR calculation method, under optimal conditions, fish and shrimp with the spiked concentrations of 0.005, 0.010, 0.020, 0.050, 0.100, 0.200, 0.500, 1.000 and 2.00 µg·L^−1^ were extracted, and the results are shown in Table 1.

##### Blank Sample Spiked Recovery and Precision

Blank samples with spiked concentrations of 1, 2, 5 and 10 µg·kg^−1^ were used for extraction experiments, where each spiked concentration was separated into six parallel groups. The recovery rate and precision of the method were investigated, and the results are shown in Table 2. The spiked recovery rate of the target in the blank fish sample was 76.31–93.26%, and RSD was 0.78–2.33%. The spiked recovery rate of the target in the blank shrimp sample was 79.53–96.26%, and RSD was 0.73–3.72%.

#### 2.4.2. Evaluation of Rapid Detection Technology

In this study, 12 samples of blank fish and 12 samples of blank shrimp were employed; three samples of each were used as the blank control, and the remaining samples were numbered according to different standard concentration levels. The sample extraction solution was obtained through sample pretreatment, an appropriate amount of oxidant 2,3-dichloro-5,6-dicyanoquinone (DDQ) was added into the standard sample to oxidize LMG to MG, and the next extraction experiment was carried out. After the eluent was air-dried to a constant volume of 1 mL, 200 μL eluent was removed and dripped onto the commercial MG rapid test strip. The results were collected, and the remaining eluent was placed on LC-MS/MS for testing. The detection results of both methods are shown in Table 3.

The results showed that the negative sample was not detected using either the instrument or the rapid test strip method, indicating that the whole experimental operation was correct and the experimental results were reliable. The results of the three blank samples at standard concentration level were consistent for the instrument and rapid test strip methods. LOD of rapid test strips was 2 μg·kg^−1^, which was determined within a short time period. However, due to the lack of suitable sample pretreatment methods, the accuracy of MG and LMG content determination decreased. In this study, MIT combined with rapid detection technology realized the accurate identification of MG and LMG by utilizing the specific selective holes on the surface of the as-prepared MG-MIM and highly selective adsorption of target objects. It had the advantages of simple, rapid, efficient and strong anti-matrix interference ability. LOD of the combined technology was 1 μg·kg^−1^, which meets the national prohibition requirements of MG.

### 2.5. Practical Sample Analysis

In order to verify the practicability of the method, 80 positive samples (including 50 fish samples and 30 shrimp samples) and 20 negative samples (including ten fish samples and ten shrimp samples) were selected. The test results of these samples were provided by the Food Inspection and Quarantine Center of Shenzhen Customs, which were detected using the standard method of GB/T 19857-2005. Then, 100 selected samples were detected using MIT as the pretreatment method, combined with MG rapid detection strip. Among them, 20 negative samples were consistent with those provided by the Food Inspection and Quarantine Center. Among 80 positive samples, the test result of one fish sample was 1.10 ng·g^−1^ MG, which was inconsistent with the test results by the proposed method. Therefore, the accuracy of the proposed method was 99%, which confirms its suitability for actual sample detection.

## 3. Experimental

### 3.1. Materials and Reagents

All chemicals were of analytical grade and used without further purification. Malachite green (MG) and leucomalachite green (LMG) were purchased from Witega Company (Berlin, Germany). Basic Red 9 and Basic Green (>98%) were purchased from Shenzhen Ester Scientific Instrument Co., Ltd. (Shenzhen, China). Methyl violet (MV, >98%) was purchased from Bide Pharmatech Ltd. (Shanghai, China). Methacrylic acid was purchased from Sigma-Aldrich Company (St. Louis, MI, USA). Moreover, 2-(Trifluoromethyl)acrylic acid (>98%) was purchased from Shanghai Anpu Experimental Technology Co., Ltd. Azobisisobutyronitrile (AIBN), and acetic acid and acrylamide (>98.5%) were purchased from Shanghai Lingfeng Chemical Reagent Company (Shanghai, China). In addition, 2-Acrylamide-2-methylpropanesulfonic acid (AMPS, >98%), toluene, hexane and chloroform were purchased from Aladdin reagent Co., Ltd. (Shanghai, China). Further, 4-Vinylbenzoic acid (>97%) was purchased from TCI (Shanghai) Development Co., Ltd. (Shanghai, China). Ethyleneglycol dimethacrylate (EGDMA) was purchased from Tokyo Chemical Industry Co., Ltd. (Tokyo, Japan). Methanol, formic acid and acetonitrile were purchased from Merck & Co., Inc. (Rahway, NJ, USA). Dimethyl sulfoxide (DMSO) was purchased from Sinopharm Chemical Reagent Co., Ltd. (Beijing, China). Sodium hypochlorite was purchased from Guangzhou Huada Chemical Reagent Co., Ltd. (Guangzhou, China).

### 3.2. Instrumentation

UPLC-MS analyses were performed using a Quadrupole Orbitrap mass spectrometer (Thermo Scientific, Waltham, MA, USA). UV–vis absorption spectra were recorded using a UV–vis spectrophotometer (Cary 300, Agilent, Palo Alto, CA, USA). FT-IR spectra were obtained using a IRAffinity-1 FT-IR spectrometer (Shimadzu, Kyoto, Japan). SEM images of MG-MIM were obtained using a JSM-5800 SEM (JEOL, Tokyo, Japan). Brunauer–Emmett–Teller (BET) specific surface areas were determined using a BET apparatus (Micromeritics, Norcross, GA, USA), with nitrogen adsorption-desorption isotherms at 77 K. The weight change of the sample was determined using a STA449F3 TGA-DSC analyzer (NETZSCH, Bavaria, Germany). The analytical atmosphere selected for this study was nitrogen, and the analyses were carried in a temperature range of 35−600 °C and at a heating rate of 10 °C min^−1^.

### 3.3. UPLC/MS Measurements

UPLC system was equipped with a C_18_ column (150 mm × 2.1 mm, i.d. = 2.7 μm), which was maintained at a temperature of 30 °C. The mobile phase was methanol with 0.1% acetic acid (solvent A) and acetonitrile (solvent B) at a flow rate of 0.25 mL·min^−1^. The injected volume was 10 μL. The mobile phase of A/B at a flow rate of 0.25 mL·min^−1^ started at 80/20 (*v*/*v*), increased to 5/95 (*v*/*v*) over a 3 min period, held for 2 min, then increased to 80/20 (*v*/*v*) over an 8 min period, and finally held for 5 min. The mass spectrometer was operated in ESI mode, with a scanning range of *m*/*z* 100–1000.

### 3.4. Preparation of Surface-Coated MIM

MIM preparation was as follows: 0.5 mmol MV, 2 mmol 4-vinylbenzoic acid, 10 mmol EGDMA, 0.5 mmol AIBN, 5 mL acetonitrile, and 0.6 mL DMSO were placed into a 100 mL heart-shaped bottle, and mixed in a water bath at 25 °C under ultrasonication. The mixed organic solution was placed in a refrigerator at 4 °C for 6 h, which allowed the combination of the template molecule and functional monomer. The mixture was removed from the refrigerator, purged with nitrogen for 5 min, and then prepolymerized at 60 °C. Then, the mixture was evenly dripped onto the glass fiber membrane, which was placed into a polytetrafluoroethylene mold, and transferred to a vacuum preservation box and evacuated. Then, it was heated in an oven at 70 °C for 16 h. The formed MIM was aged at 120 °C for 2 h, and cleaned with high-purity water, acetic acid/acetonitrile (1:9, *v*/*v*), and acetonitrile, respectively. The synthesized MIM material was oxidized at 80 °C in a beaker with sodium hypochlorite solution until the material turned white or light yellow, and repeatedly washed with high-purity water to remove the remaining sodium hypochlorite solution. The preparation of NIM was conducted under the reaction conditions for MIM preparation except for template molecule.

### 3.5. Adsorption Experiments

#### 3.5.1. Static Adsorption Experiment

An appropriate amount of mixed standard solution (two substances in total, MG and LMG) was placed in 16 polyvinyl chloride centrifuge tubes, and mixed with 20% acetonitrile/PBS buffered salt solution (PBS buffered salt solution pH = 7.2, *v*/*v*), generating a series of mixed standard solutions of different concentrations (0–30 ng·mL^−1^). Then, 16 pieces of MIM (size 2 × 7 cm) of the same specification were placed into the above 16 polyvinyl chloride centrifuge tubes in sequence. The tubes were continuously shaken in the oscillator for 20 min at a shaking rate of 60 times/min. Then, MG-MIM was removed and transferred to Ziploc bags. After addition of 1 mL acetic acid/acetonitrile (*v*/*v* = 1:9) solution, the Ziploc bags were sealed and ultrasonicated for 5 min. The liquid in the Ziploc bags was removed and placed into the sample injection vial, and its volume was made up to 1 mL with high-purity water. The sample was analyzed with UPLC-MS/MS. The extraction operation of NIM was conducted under the same conditions.

#### 3.5.2. The Experiment of Adsorption Kinetics

An appropriate amount of mixed standard solution (two substances in total, MG and LMG) was placed in 19 polyvinyl chloride centrifuge tubes, and mixed with 20% acetonitrile/PBS buffered salt solution (pH PBS buffered salt solution = 7.2, *v*/*v*) generating a series of mixed standard solutions with a concentration of 20 ng·mL^−1^. Then, 19 pieces of MIM (size 2 cm × 7 cm) of the same specification were placed into the tubes in sequence and shaken in an oscillator at a rate of 60 rpm·min^−1^ for 0 min, 0.5 min, 1 min, 1.5 min, 2 min, 2.5 min, 3 min, 3.5 min, 4 min, 5 min, 10 min, 15 min, 20 min, 30 min, 40 min, 60 min, 80 min, 100 min, and 120 min. Then, MIM samples were removed and placed into Ziploc bags. After the addition of 1 mL acetic acid/acetonitrile (*v*/*v* = 1:9) solution, the Ziploc bags were sealed and ultrasonicated for 5 min. The liquid in the Ziploc bags was removed and placed into the sample injection vial, and its volume was made up to 1 mL with high-purity water. The sample was analyzed with UPLC-MS/MS. The extraction operation of NIM was conducted under the same conditions.

The experimental data were fitted using pseudo-first-order and pseudo-second-order kinetic models as follows: [25,26]

Pseudo-first-order kinetic model equation:(1)$$\ln\left(Q_{e}-Q_{t}\right)=\ln Q_{e}-k_{1}t$$

Pseudo-second-order kinetic model equation:(2)$$\frac{t}{Q_{t}}=\frac{1}{k_{2}Q_{e}{}^{2}}+\frac{t}{Q_{e}}$$
where *Q_e_* (ng·cm−^2^) and *Q_t_* (ng·cm^−2^) are the adsorption capacities at equilibrium and any time *t* (min), respectively; *k*_1_ is the rate constant of the pseudo-first-order model, and *k*_2_ is the rate constant of the pseudo-second-order model.

### 3.6. Sample Pretreatment

About 5 g fish sample was added to 1.4 mL of 0.1 mol·L^−1^ sodium acetate (pH = 4.8) and 0.6 mL of 1 mol·L^−1^
*p*-toluenesulfonic acid. The mixture was vortexed for 30 s, shaken, and extracted using 10 mL acetonitrile for 5 min. Then, 2 g anhydrous magnesium sulfate and 4.5 g neutral alumina were added, followed by vortexing for 30 s. The mixture was centrifuged at 9500 rpm for 5 min in a low-temperature centrifuge, and all the supernatant was removed and passed through an organic filter membrane (0.22 μm).

## 4. Conclusions

In conclusion, the surface molecular imprinting method was used for the preparation of MG-MIM utilizing MV as the target structural analog, the substitute template molecule, 4-vinylbenzoic acid as the functional monomer, EGDMA as the crosslinking agent, and acetonitrile as the polymerization solvent. Characterization was conducted using SEM, BET, FT-IR and TGA-DSC analyses. The results showed that MG-MIM possessed a fluffy surface, porous and loose structure, and good thermal stability, which were conducive to the adsorption of the target. MG-MIM had a high adsorption rate for both MG and LMG, and the adsorption equilibrium was reached in 20 min. The adsorption conformed to the quasi-second-order kinetic model, mainly chemical adsorption. MG-MIM adsorption, combined with rapid detection method, accurately detected MG and LMG content, with a linear range of 0–50 µg·L−1, linear correlation coefficients greater than 0.9991, and LODs of 0.005–0.02 µg ·kg−1, which conformed to the national prohibition requirements of MG. The combination of molecular imprinting technology and rapid detection technology was suitable for the detection of actual samples, providing a novel method for the detection of trace MG in a complex matrix.

## Figures and Tables

**Figure 1 molecules-28-00310-f001:**
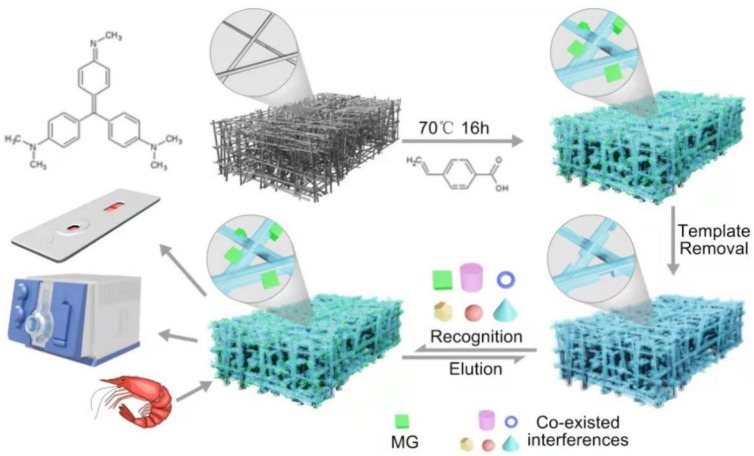
Schematic diagram of surface-coated MIM preparation and analytical procedures.

**Figure 2 molecules-28-00310-f002:**
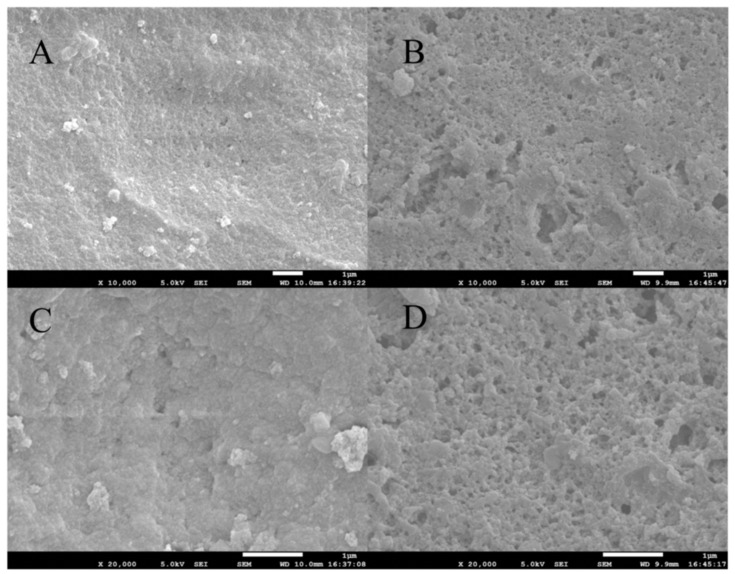
SEM images of (**A**) MG-MIM and (**B**) NIM at 10 K, and images of (**C**) MG-MIM and (**D**) NIM at 20 K.

**Figure 3 molecules-28-00310-f003:**
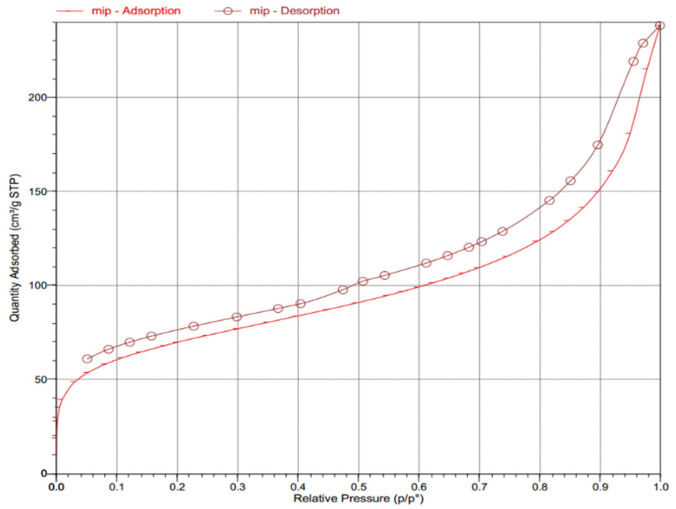
Nitrogen adsorption–desorption isotherms of MG-MIM.

**Figure 4 molecules-28-00310-f004:**
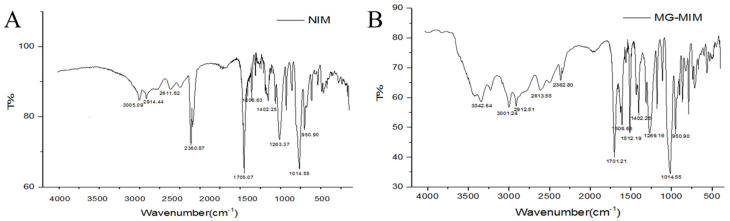
FT-IR spectra of NIM (**A**) and MG-MIM (**B**).

**Figure 5 molecules-28-00310-f005:**
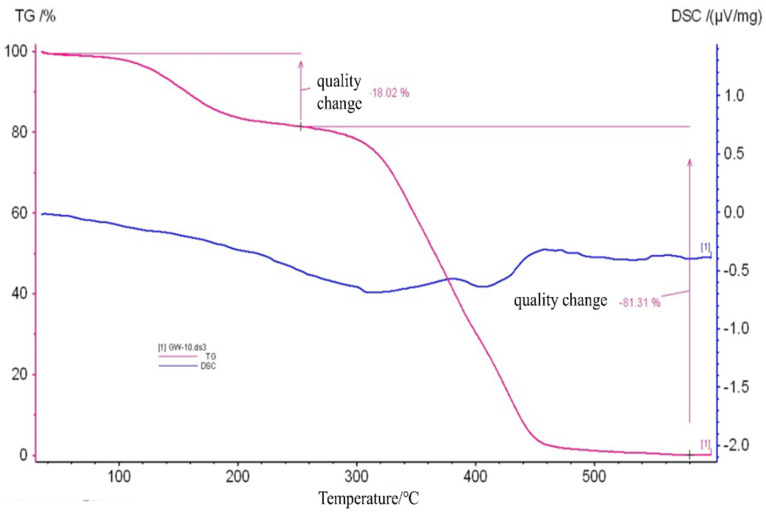
TGA/DSC curves of MG-MIM.

**Figure 6 molecules-28-00310-f006:**
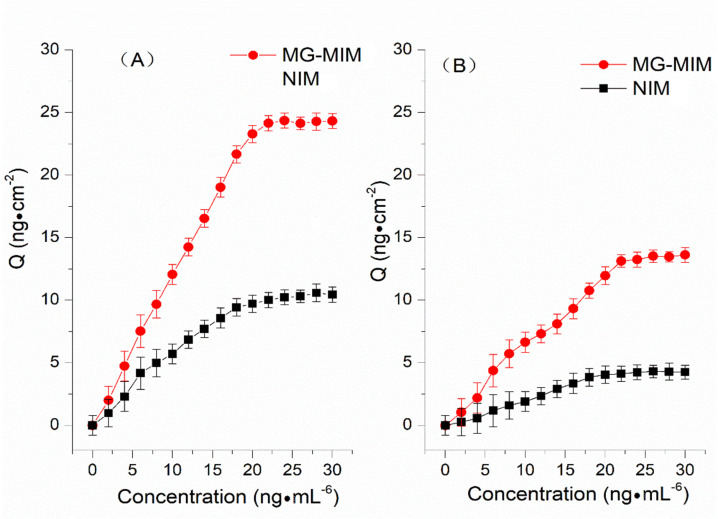
Adsorption isotherm curves of MG (**A**) and LMG (**B**).

**Figure 7 molecules-28-00310-f007:**
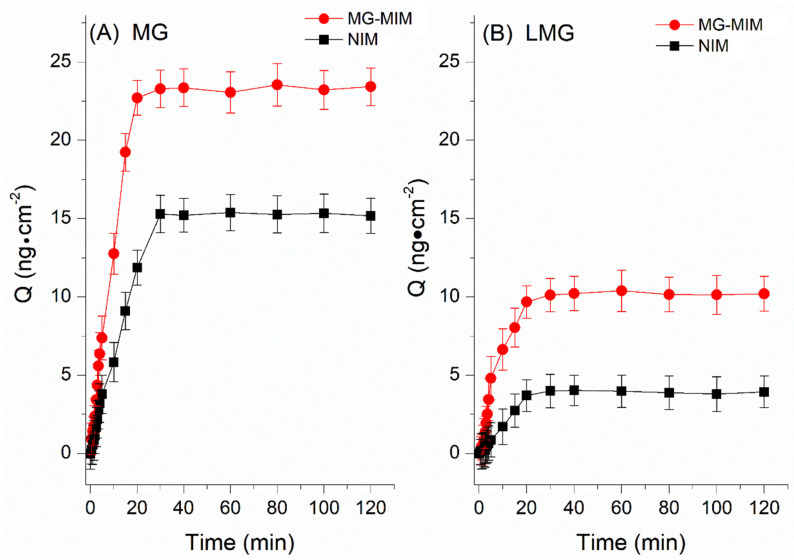
Adsorption dynamic curves of (**A**) MG and (**B**) LMG.

**Figure 8 molecules-28-00310-f008:**
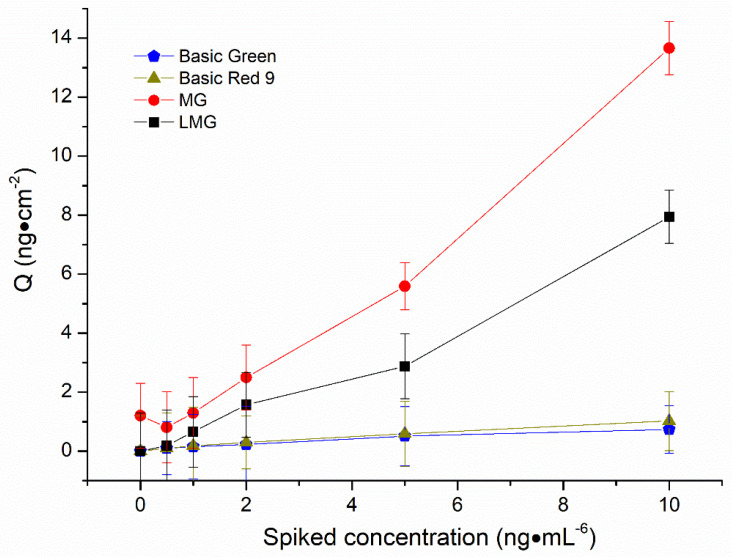
Comparison of MG-MIM extraction for basic green, basic red 9, MG and LMG at the concentration of 10.0 ng.mL^−6.^.

**Figure 9 molecules-28-00310-f009:**
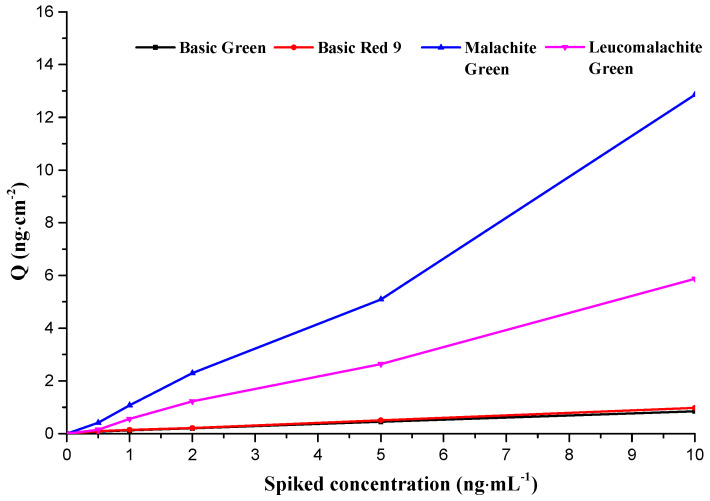
Adsorption curves of MG-MIM for different substances.

**Table 1 molecules-28-00310-t001:** Regression equations, LODs and LOQs of MG and LMG.

Matrix	Compound	Regression Equations	Correlation Coefficients (R^2^)	LOD (µg·kg^−1^)	LOQ (µg·kg^−1^)
Standard solution	MG	Y = −5.43901e + 006 + 1.64419e + 007X	0.9998	0.005	0.020
	LMG	Y = 6.44322e + 007 + 5.53649e + 007X	0.9994	0.005	0.010
Fish	MG	Y = 1.74e^6^ + 3.51e^7^X	0.9997	0.005	0.020
	LMG	Y = 2.36e^6^ + 2.32e^7^X	0.9995	0.020	0.050
Shrimp	MG	Y = −2.55e^6^ + 2.67e^7^X	0.9993	0.005	0.020
	LMG	Y = 4.42e^6^ + 5.94e^7^X	0.9991	0.020	0.050

**Table 2 molecules-28-00310-t002:** Results of the recovery and accuracy evaluation experiments in spiked samples.

Sample	Target	1 µg·kg^−1^		2 µg·kg^−1^		5 µg·kg^−1^		10 µg·kg^−1^	
		Recovery%	RSD%	Recovery%	RSD%	Recovery%	RSD%	Recovery%	RSD%
Fish	MG	76.31	1.49	87.25	0.96	86.02	1.12	81.85	1.30
	LMG	80.75	1.28	93.26	0.78	85.65	2.18	82.04	2.33
Shrimp	MG	85.35	0.79	89.16	2.76	96.26	0.70	84.69	0.73
	LMG	79.53	3.62	93.32	1.73	86.12	3.72	82.70	2.43

**Table 3 molecules-28-00310-t003:** Results of LC-MS/MS and quick test strip in spiked sample.

Sample	Method	Blank			The Standard Concentration Level								
					1.0 ng·g^−1^			2.0 ng·g^−1^			5.0 ng·g^−1^		
Fish	C (ng·mL^−1^)	0.000	0.010	0.010	0.843	1.183	0.858	1.868	1.806	1.720	4.527	4.696	4.491
	Quick test strip	−	−	−	+	+	+	+	+	+	+	+	+
Shrimp	C (ng·mL^−1^)	0.026	NF	0.021	0.815	1.036	1.052	1.784	1.741	1.850	4.849	4.775	4.759
	Quick test strip	−	−	−	+	+	+	+	+	+	+	+	+

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
