# Peer review of "Dummy Template-Based Molecularly Imprinted Membrane Coating for Rapid Analysis of Malachite Green and Its Metabolic Intermediates in Shrimp and Fish"

_molecules, 2022, doi:10.3390/molecules28010310_

Round 1
Reviewer 1 Report
In this study, A novel malachite green molecularly imprinted membranes (MG-MIMs) with specific selectivity for malachite green (MG) and leucomalachite green (LMG) was prepared using a hydrophobic glass fiber membrane as the polymer substrate, methyl violet as a template analogue, 4-vinybenzoic acid as the functional monomer, and ethyleneglycol dimethacrylate as the crosslinking agent. The specific recognition ability to the targets was good and the adsorption capacity was stable even after 5 cycles. MG and LMG were determined in shrimp and fish with low matrix effect, with good linear correlation and high detection sensitivity. However, the manuscript should be addressed following issues and revised in detail.
1. In the abstract, said that malachite green molecularly imprinted membranes (MG-MIMs) with good linear correlation and high detection sensitivity, so the linear relationships and sensitivity values should be reflected in the abstract.
2. There are some writing errors in L77.
3. L229, “Compared with MIMs, MG-MIMs had…”, it should be “Compared with NIMs, …”.
4. There are two Figure1 on Page 3 and Page 6. On Page 6, Fig. 1A may be confused with 1B, and Fig. 1C may be confused with 1D, which must be checked carefully.
5. The surface structure and morphological characteristics of the as-prepared MIMs also should be examined with scanning electron microscope (SEM), which is very important for detecting the trace MG in complex matrix.
Reviewer 2 Report
Authors presented application of a glass fiber membrane was employed as the substrate and methyl violet (MV) as the dummy template molecule, allowing the synthesis of MG-MIMs by surface molecular imprinting. Through the combination of MIT with rapid detection technology, MIMs exhibited accurate target recognition, and effectively improved inaccurate detection analytes.
The publication is prepared in accordance with the traditional convention concerning the synthesis and application of MIMs. In this respect, it is not innovative. The paper contains all the information on the method of obtaining the material for its identification and research on the thermodynamics and kinetics of the binding process of selected analytes. The obtained results indicate the possibility of using this type of material for the analysis of selected analytes. Authors presented application of a glass fiber membrane was employed as the substrate and methyl violet (MV) as the dummy template molecule, allowing the synthesis of MG-MIMs by surface molecular imprinting. Through the combination of MIT with rapid detection technology, MIMs exhibited accurate target recognition, and effectively improved inaccurate detection analytes.
The publication is prepared in accordance with the traditional convention concerning the synthesis and application of MIMs. In this respect, it is not innovative. The paper contains all the information on the method of obtaining the material for its identification and research on the thermodynamics and kinetics of the binding process of selected analytes. The obtained results indicate the possibility of using this type of material for the analysis of selected analytes. System was used to rapid detection analytes. Chemistry obtained a novel method for the detection of trace MG in complex matrix.
The prepared publication is another work showing the universality of MIP and MIM methods in the selective analysis of selected analytes, it already extends the extensive knowledge in this field.
Reviewer 3 Report
This is an interesting research related to the preparation of molecularly imprinted membranes. However, I cannot recommend the paper in the submitted form for the publication. It must be revised before the final acceptation.
The detailed comments:
1. Abstract: ..”specific recognition ability to the targets was good” – what does it mean „good”? Please specify it.
2. Line 77-78: “MIMs 是 s 77 s owing” – what does it mean? Please correct it.
3. IR spectrum (Fig. 3) – please, remove the initial section with a lot of noise or apply a smoothing effect
4. How durable was the connection of the MIP polymer layer with the membrane. Didn't the weight of the graft layer decrease during the process as a result of washing out?
5. Is it possible to determine the model according to which sorption occurs for these membranes?
6. Was the contact time of the template - imprint checked?
7. How did the porosity of the membranes change after the polymerisation? What kind of membranes were used? Was the water flux checked before and after modification? If not, please do it.
Round 2
Reviewer 3 Report
Thank you for the corrections.